# Simultaneous Lateral Flow Immunoassay for Multi-Class Chemical Contaminants in Maize and Peanut with One-Stop Sample Preparation

**DOI:** 10.3390/toxins11010056

**Published:** 2019-01-20

**Authors:** Du Wang, Jianguo Zhu, Zhaowei Zhang, Qi Zhang, Wen Zhang, Li Yu, Jun Jiang, Xiaomei Chen, Xuefang Wang, Peiwu Li

**Affiliations:** 1Oil Crops Research Institute of the Chinese Academy of Agricultural Sciences, Wuhan 430062, China; wangdu@caas.cn (D.W.); jianguozhu01@126.com (J.Z.); yuli01@caas.cn (L.Y.); jiangjun@caas.cn (J.J.); chenxiaomei@caas.cn (X.C.); wangxuefang01@caas.cn (X.W.); 2Quality Inspection and Test Center for Oilseeds Products, Ministry of Agriculture and Rural Affairs, Wuhan 430062, China; 3Key Laboratory of Biology and Genetic Improvement of Oil Crops, Ministry of Agriculture and Rural Affairs, Wuhan 430062, China; 4Key Laboratory of Detection for Mycotoxins, Ministry of Agriculture and Rural Affairs, Wuhan 430062, China; 5Laboratory of Risk Assessment for Oilseeds Products (Wuhan), Ministry of Agriculture and Rural Affairs, Wuhan 430062, China

**Keywords:** simultaneous lateral flow immunoassay, multi-class chemical contaminants, aflatoxin B_1_, zearalenone, chlorothalonil, maize and peanut

## Abstract

Multi-class chemical contaminants, such as pesticides and mycotoxins, are recognized as the major risk factors in agro products. It is thus necessary to develop rapid and simple sensing methods to fulfill the on-site monitoring of multi-class chemical contaminants with different physicochemical properties. Herein, a lateral flow immunoassay via time-resolved fluorescence was developed for the rapid, on-site, simultaneous, and quantitative sensing aflatoxin B_1_ (AFB_1_), zearalenone (ZEA), and chlorothalonil (CTN) in maize and peanut. The sample preparation was optimized to a single step, combining the grinding and extraction. Under optimal conditions, the sensing method lowered the limits of detection (LOD) to 0.16, 0.52, and 1.21 µg/kg in maize and 0.18, 0.57, and 1.47 µg/kg in peanut with an analytical range of 0.48–20, 1.56–200, and 3.63–300 µg/kg for AFB_1_, ZEA and CTN, respectively. The protocol could be completed within 15 min, including sample preparation and lateral flow immunoassay. The recovery range was 83.24–110.80%. An excellent correlation was observed between this approach and high-performance liquid chromatography-tandem mass spectrometry (HPLC-MS/MS) for mycotoxins and gas chromatography-tandem mass spectrometry (GC-MS/MS) for pesticide in maize and peanut. This work could be applied in on-site multi-class sensing for food safety.

## 1. Introduction

Agro products contain various hazardous compounds, in particularly mycotoxins and pesticides [1,2,3]. They can occur throughout the entire process of agro production, from the farm to table [4]. Fungicides are commonly used to control adverse effects from fungal diseases, such as *A. flavus*, *A. parasiticus*, *F. graminearum*, *F. culmorum* and *F. crookwellense*, in agricultural practices, affecting the nervous system, cancer, and causing acute death [5]. Chlorothalonil (CTN), which is a broad-spectrum and protective organic chlorine bactericide is, especially in developing countries, the most widely used fungicide for controlling mold, mildew, and bacteria [6,7,8]. On the other hand, mycotoxins are mainly produced by fungi, and are carcinogenic, teratogenic, nephrotoxic, and mutagenic [9]. Among the mycotoxins, aflatoxin B_1_ (AFB_1_) and zearalenone (ZEA) are frequently and widely found in agricultural products [10]. AFB_1_, a Group 1 carcinogen classified by the International Agency for Research on Cancer, can lead to liver cirrhosis, carcinomas, and lethality in humans and livestock [11]. ZEA is relatively low in acute and chronic toxicity, carcinogenicity, and genotoxicity, and can induce DNA-adduct formation [12]. Although fungicides are supposed to protect the agro products from fungi as well as mycotoxins, fungicides and mycotoxins have usually been found to co-occur. The accumulating evidence demonstrates the co-existence of AFB_1_, ZEA and CTN in agro products [13].

Owing to their extreme toxicity and universal co-existence, CTN and mycotoxins are an ever-increasing concern in food safety. Many countries and international organizations have set the maximum residue limits (MRLs) for AFB_1_, ZEA, and CTN, respectively. The European Commission has set the MRLs as 4 and 2 μg/kg for the sum of aflatoxins (AFB_1_, AFB_2_, AFG_1_, and AFG_2_) and AFB_1_ in cereals, respectively. The MRL for ZEA is 60 μg/kg in unprocessed maize; AFB_1_ is 20 μg/kg, while those for CTN are 50 and 100 μg/kg in peanut and maize, respectively. Given the potential role of multiple contaminants as prognostic markers, it is essential to establish a rapid, simultaneous, and quantitative sensing method for monitoring the co-occurrence of the AFB_1_, ZEA and CTN.

Various methods have been introduced for the detection of AFB_1_, ZEA and CTN, including gas chromatography (GC), high-performance liquid chromatography (HPLC), HPLC-tandem mass spectrometry (HPLC–MS/MS) [14], and enzyme-linked immunosorbent assay (ELISA) [15]. It is a recent trend in analytical chemistry to develop rapid and simple detection methods for processing analytes of multiple classes of chemical contaminants with different physicochemical properties. There have been several multi-class methods for detection of chemical contaminants, usually based on HPLC-MS/MS [16,17,18,19]. These analytical methods have many advantages, such as excellent quantitation, precision, and accuracy. However, they have the disadvantages of requiring excessing time and labor, and expensive equipment, well-trained personnel, and prolonged operating time. These methods are thus restricted to use in the laboratory and are not adaptable to on-site application for food safety.

Fortunately, the lateral flow immunoassay has been developed in response to the need for on-site rapid sensing. It has been successfully applied in biological, medicine, food, and environmental analysis [20,21,22,23]. The sensing has the advantages of rapidness, ease of use, cost-efficiency, portability, sensitivity, and specificity. The lateral flow immunoassay systems use a variety of labels, including gold, silver, fluorescent and electrochemical/magnetic particles. Gold is the most popular immunochromatographic label. However, the gold label is susceptible to interference in complex backgrounds caused by matrices in food. Therefore, the elimination of background interference is essential for highly sensitive sensing [24,25,26,27]. The fluorescent lateral flow immunoassay, based on lanthanide Eu^3+^ chelate labeling with unique fluorescence properties, boasts a prolonged fluorescent life (the order of ms), large Stokes shift, narrow-band emission lines, and high quantum yields. These unique advantages can decrease noise from background signals and scattered light [28,29,30,31], thereby resulting in enhanced sensitivity by recording the fluorescent signal from ns to μs-long lifetimes. This method was developed to detect AFB_1_ more sensitively than radioimmunoassay or ELISA methods [30,32].

When facing multiple targets of interest (mycotoxins and pesticides), the fabrication of multiple lateral flow immunoassays and sample preparations remains the major challenge. Herein, we propose an on-site, rapid, simultaneous, and quantitative lateral flow immunoassay for AFB_1_, ZEA, and CTN with three monoclonal antibodies. After optimization, this protocol was validated and evaluated. Subsequently, the comparison between this proposal and HPLC-MS/MS or GC-MS/MS was conducted. Finally, this lateral flow immunoassay was used to sense AFB_1_, ZEA, and CTN in real maize and peanut samples. The present study allows an alternative for synchronously sensing the multiple kinds of targets, especially for the on-site purpose of ensuring food safety.

## 2. Results and Discussion

### 2.1. Preparation mAb@Eu-Nanosphere Particle Conjugates

Carboxylated Eu-nanosphere particles served as the carrier for mAb conjugation. When the pH was slightly higher than the isoelectric point of mAb, the conjugation of mAb on Eu-nanosphere particles was easy. The optimal pH value for mAb@Eu-nanosphere particle conjugation was 8.2, 8.0, and 8.2 for anti-AFB_1_, anti-ZEA, anti-CTN, respectively. The dosage of antibodies was optimized to achieve maximal sensitivity and stability. The optimal dose was prepared with different mass ratios of the mAb and Eu-nanosphere particles (0, 5, 10, 20, 30, 40, 50, 60, 70, 80, and 100 µg/kg). To prevent mAb@Eu-nanosphere particles from aggregating, the lowest stabilizing mAb concentration was optimized, by determining the protein absorbance of the supernatant after conjugation. The optimal mAb concentrations of the three mAb@Eu-nanosphere particle conjugates were 10, 30, and 80 µg/mL for AFB_1_, ZEA, and CTN, respectively.

### 2.2. Optimization of Lateral Flow Strip

The lateral flow immunoassay provided a single-run detection method based on the competitive immunoassay format. The sensing quantification was dependent on the fluorescence intensities of T and C lines, while the sensitivity relied on the sensitivity of the mAb and the concentrations of antigens on the T lines.

To obtain superior sensitivity, experiments were used to determine the optimal immunoreagent concentrations and dilution factors of the mAb@Eu-nanosphere particle conjugates by using both blank solution and standard solution (0.25, 2.5, 10.0 µg/kg for AFB_1_, ZEA, CTN).

The serial diluted ratio of mAb@Eu-nanosphere particle solutions were used to obtain a minimum dosage of mAb on the clear T lines use protective reagent. The optimal concentration, dosage, and dilution ratio for the labeled anti-AFB_1_, anti-ZEA, and anti-CTN mAbs were evaluated based on ten repetitions, respectively. The T lines displayed a clear color, when the mAb concentrations were 10, 30, and 80 µg/mL with the mAb@Eu-nanosphere particle dilution of 400×, 300×, and 300× for AFB_1_, ZEA, and CTN, respectively. These results were in agreement with the recognition that the mAbs with a high affinity had a reduced consumption [33].

Furthermore, various factors can influence the performance of lateral flow immunoassay, such as the amount of immunoreagent on the T lines, the type of nitrocellulose membrane, and the blocking buffer for the sample pad. In addition, 0.6 µL/cm AFB_2a_-BSA (0.25 mg/mL), 0.7 µL/cm ZEA-BSA (0.5 mg/mL), 0.7 µL/cm CTN-BSA (4 mg/mL), and 0.6 µL/cm rabbit anti-mouse IgG (0.25 mg/mL) were optimized to dispense on the T and C lines, respectively.

### 2.3. Evaluation of the Lateral Flow Strip Method

#### 2.3.1. Calibration Curve

To establish a standard curve, the spiked sample and the standard solution were employed, respectively. For negative controls, the mAb@Eu-nanosphere particles present in the solution flowed laterally along the strip towards the T lines where the mAb@Eu-nanosphere particle complexes are captured by the immobilized antigens on the T lines [T_(CTN)_, T_(AFB1)_, T_(ZEA)_]. The excess mAb@Eu-nanosphere particles flowed towards the C line, and AFB_1_/ZEA/CTN-mAb@Eu-nanosphere particle conjugates were captured by the rabbit anti-mouse IgG on the C line. In the case of positive samples, AFB_1_/ZEA/CTN-conjugated mAb@Eu-nanosphere particles did not interact with the immobilized complex on the T lines, but reached the C line, which was indicated by a faint red color (i.e., a positive sample with a low concentration of the analyte) or no color (i.e., a positive sample with a high concentration of the analyte) on the T lines (Figure 1B,C). Then, the T and C lines were inspected by the multi-target reader. This showed that the fluorescent intensity ratio was correlated with the concentration and the three T lines did not interfere with each other. The fluorescence intensity ratios on the T and C lines were inversely proportional to the analyzed concentrations. Therefore, the levels of the analytes were determined in the samples.

The plot of the ratio of the T line value to that of the C line vs. the natural logarithm of the concentration (ln c), established a correlation, which was dependent on the different matrices: Y = a·log [c] + b. The series of concentrations for the analysis of AFB_1_ (0, 1, 2, 10, 20 µg/kg), ZEA (0, 10, 20, 100, 200 µg/kg), and CTN (0, 10, 50, 100, 300 µg/kg) are shown in Figure 2, and the equations and correlation coefficient (R^2^) are shown in Table 1. The analytical range of the sensing method was 0.48–20, 1.56–200, and 3.63–300 µg/kg for AFB_1_, ZEA, and CTN, respectively.

#### 2.3.2. Sensitivity

Both blank solutions and sample extracts containing different concentrations of analytes were prepared and analyzed by lateral flow immunoassay. The standard deviations were calculated to evaluate the LOD of analytical method independently. After the standard deviations (SDs) were obtained from twenty-one measurements of blank matrices of different samples, each LOD was calculated as 3-fold of the SD divided by the absolute slope of the standard curve. The LOQ was calculated as 3-fold of the LOD, representing the lower limit of the linear range. The upper limit of the linear range depended on the color-fading of the T lines. The lower LOD was obtained from ten repetitions. The above two limit values were used as the critical linear range of the calibration curves. According to the above equation, the LODs for ZEA, AFB_1_, and CTN were calculated (Table 1). The sensing method lowered the limits of detection (LOD) to 0.16, 0.52, and 1.21 µg/kg in maize and 0.18, 0.57, and 1.47 µg/kg in peanut for AFB_1_, ZEA, and CTN along with the linear ranges from the calibration curves. This established method had an evidently lower LOD when compared with the LOD via the nanogold-based strip method (0.25 μg/kg, 1μg/kg for AFB_1_, ZEA, 5μg/kg for CTN) [10,34,35].

#### 2.3.3. Repeatability and Reproducibility

Maize and peanut were chosen as the experimental matrixes. When standard addition recovery experiments were conducted, three different concentration levels of each analyte were selected within the linear range of the calibration curves after processing the conversion. Then, these concentrations were spiked into negative maize AFB_1_ (1, 5, 10 µg/kg), ZEA (5, 20, 60 µg/kg), CTN (10, 20, 50 µg/kg) were investigated, followed by preparation and detection with lateral flow immunoassay. The accuracy of lateral flow immunoassay was mainly evaluated by the average recovery from five parallelly repeated measurements. The precision was calculated by the relative standard deviation (RSD) of the above measurements spiked with different concentration levels. Additionally, the stability of five intra-day measurements was compared with that of inter-day (*n* = 5) for reproducibility. The average recovery for reproducibility was 91.57% (83.24–110.80%), coefficient of variation (CV) of 5.0% (1.7–9.1%). For intraday reproducibility, the test strips of five different days were used for replicate testing of the samples, and the average recoveries were 91.48% (83.52–108.25%), with a mean CV of 4.8% (0.9–6.3%) (Table 2).

### 2.4. Sample Preparation

Apart from the optimization of the sensing method, one major challenge lies in the sample preparation, especially for multi-class targets of interest. It would be easier for a single target or single class, because the targets pose similar physical properties during the extraction. To realize the successful sensing of multi-class targets, a one-stop sample preparation should be developed [16,36]. Then, the extraction requests complex clean-up, thus prolonging the sensing time span. Among the glass fiber, polyester, nitrocellulose and cellulose, the GE Fusion 5 was optimized using the sample pad. This pad can efficiently reduce the matrix, thus enhancing the sensitivity. The one-stop sample preparation, including sample grinding and extraction, was simple and required only a short time, compared with the existing sample extraction for multiple classes using HPLC (GC)-MS/MS methods [18,19,20].

### 2.5. Determination and Evaluation of Real Samples

The eight random naturally contaminated maize and peanut samples were analyzed by the developed lateral flow immunoassay in order to detect AFB_1_, ZEA, and CTN, followed by other large, precise instruments for confirmation. In this comparison test, HPLC-MS/MS was selected as the confirmed detection method for the simultaneous analysis of AFB_1_ and ZEA, while GC-MS/MS was the optimum choice for CTN. Each sample was prepared and detected three times, and all of the test results were investigated after data processing. Notably, the reaction solution beyond the scope of the linear range in the application of lateral flow immunoassay requires dilution to a suitable concentration level, followed by detection with a lateral flow immunoassay. The lateral flow immunoassay result had a better correlation with the precision measurement method; R^2^ > 0.88. As shown in Table 3, AFB_1_ were found in 62.5% of the 16 samples (4 of 8 maize, 6 of 8 peanut), ZEA were found in 31.25% of the 16 samples (5 of 8 maize), and CTN were found in 37.5% of the 16 samples (3 of 8 maize, 3 of 8 peanut). The consistency among these methods indicated that the lateral flow immunoassay was able to satisfy the requirements of rapid screening on site and even entire quantitation of the multiple components simultaneously.

## 3. Conclusions

A novel immunochromatographic assay based on time-resolved fluorescent immunochromatography was developed that successfully allowed the rapid and simultaneous detection of AFB_1_, ZEA, and CTN in maize and peanut. This approach can simultaneously and quantitatively detect mycotoxins and pesticides. After optimization, the prepared strip was lowered the LOD of 0.16, 0.52, and 1.21 µg/kg in maize and 0.18, 0.57, and 1.47 µg/kg in peanut with an analytical range of 0.48–20, 1.56–200, and 3.63–300 µg/kg for AFB_1_, ZEA, and CTN, respectively. The time required to perform the analysis was 15 min, including sample preparation. The recovery ranged from 83.24% to 110.80%. The intra- and inter-day evaluations were also investigated, thereby suggesting that the method was stable and reliable. An optimal correlation between this approach and HPLC-MS/MS for mycotoxins and GC-MS/MS for pesticides was observed in peanut and corn samples, respectively.

## 4. Materials and Methods

### 4.1. Reagents and Apparatus

Standard solutions of AFB_1_, ZEA, CTN, and AFB_2a_-BSA (bovine serum albumin) were purchased from Sigma-Aldrich (Urbana, IL, USA). ZEA-BSA was purchased from Aokin AG (Berlin, Germany). The Anti-AFB_1_ monoclonal antibody and Anti-ZEA monoclonal antibody were prepared in mouse ascites fluid according to previously reported methods in our laboratory. CTN-BSA and CTN antibody were obtained from Wuxi Determine Biotech Co., Ltd. (Wuxi, China). The anti-mouse IgG was purchased from Boster Bioengineering Co., Ltd. (Wuhan, China). Mannitol, 1-Ethyl-3-(3-dimethylaminopropyl) carbodiimide (EDC), sucrose, polyvinylpyrrolidone (PVP), and Tween-20 were purchased from Sinopharm Chemical Reagent Co., Ltd. (Shanghai, China). Eu-nanosphere particles Eu-nanosphere particles were obtained from Shanghai Uni-bio Biological Co. (190 ± 10 nm particle size, solid content 0.94%). The nitrocellulose membranes and absorbent pads were procured from Millipore (Bedford, OH, USA). The XYZ 3050 Biostrip Dispenser and CM 4000 Cutter (Bio-Dot, Irvine, CA, USA) were used to prepare lateral flow immunoassay.

### 4.2. The Portable Multi-Target Reader

The portable multi-target reader was modified for sensing the multiple targets, quantitatively, from the home-made reader [30,37]. The excitation light was 365 nm via the light emitting diode (LED), while the signal acquisition was obtained at 613 ± 10 nm using a photodiode array. The signal data were processed by smoothening and derivation. The operation software included the multi-region analysis and computation module. The fluorescence intensities on both test line (T lines) and control line (C line) were recorded and calculated using data processing software for quantitative analysis.

### 4.3. Synthesis mAb@Eu-Nanosphere Particle Conjugates

The mAb@Eu-nanosphere particles were prepared as described previously [28,30]. Briefly, after 200 μL Eu-nanosphere particles were suspended in 800 μL boric acid buffer solution, 40 μL EDC (15 mg/mL) was added in one batch. The mixture was stirred for 15 min, and the suspension separated by centrifugation at 17,000× *g* for 10 min. The precipitate was re-suspended in 1 mL boric acid buffer by ultrasonication for 1 min. After three mAbs (AFB_1_, ZEA, and CTN) were added separately at 10 serial concentrations, the reaction was agitated for 12 h before centrifugation at 17,000× *g* for 10 min. The residue was re-suspended in 1 mL boric acid buffer (containing 0.5% BSA), and the reaction continued for an additional 2 h under shaking at 20 °C. Subsequently, after centrifugation at 17,000× *g* for 10 min, the precipitate was resuspended in 1 mL boric acid buffer. The mAb@Eu-nanosphere particles was diluted at a suitable ratio in the protective reagent before lyophilization in the microwell, followed by preservation at 4 °C for further experiments.

### 4.4. Fabrication of Lateral Flow Immunoassay

The lateral flow immunoassay is shown in Figure 1A. The sample pad, nitrocellulose membrane, and absorbent pad were assembled sequentially onto a plastic backing sheet. Briefly, three antigens (AFB_2a_-BSA, ZEA-BSA, and CTN-BSA) and goat anti-mouse IgG antibody were coated on the NC membrane (HF07502S25, Millipore, Bedford, OH, USA, 25 mm × 300 mm) as T_CTN_ line, T_AFB1_ line, and T_ZEA_ line) and C line using the XYZ 3050 biostrip dispenser. After overnight drying at 37 °C, the NC membrane was sealed in a plastic bag and stored under dry conditions at room temperature. The C line was located 5 mm distal to the T lines on the NC membrane. The nitrocellulose membrane, sample pad, and absorbent pad were assembled sequentially with 1–2 mm overlap of each component. The whole assembled scale board was divided into 4.5 mm × 70 mm strips with a guillotine cutter and stored at 4 °C. The strips and microwell conjugate mixtures were housed in a plastic cassette with silica desiccant gel and stored under dry conditions.

### 4.5. Principle of Lateral Flow Immunoassay

The simultaneous lateral flow immunoassay quantification was based on the competitive inhibitory interaction. The mAb@Eu-nanosphere particle conjugates were diluted by 50×, 100×, 200×, 300×, and 500× with protective reagents (2.5% sucrose, 1% BSA, and 1% mannitol in 0.01 M PBS buffer at pH 7.4). The 0.25 µg/kg of AFB_1_, 2.5 µg/kg of ZEA, and 10.0 µg/kg of CTN standard solutions and blank solutions were utilized to determine the optimum condition. After lyophilization, the 150 µL solution was injected into microwell mixed for 5 s. Then, the strip was inserted into the microwell and incubated at 37 °C for 8 min. Subsequently, the portable multi-target reader was introduced to record the fluorescent signal on the C line and three T lines.

### 4.6. Validation of Lateral Flow Immunoassay

The calibration curve, sensitivity, recovery, specificity, accuracy, and stability were investigated using the blank peanut and maize samples spiked with a series of concentrations of mycotoxins and pesticide. Recovery was obtained by spiking the sample with three concentrations of AFB_1_, ZEA, and CTN, respectively. The specificity was evaluated by the experiment using the spiked sample with over-loading concentration interference. The accuracy and stability were obtained by the intra-assay and inter-assay, respectively.

### 4.7. Comparison between Lateral Flow Immunoassay and HPLC-MS/MS (GC-MS/MS)

The samples were purchased from the local farm/market to estimate the reliability of lateral flow immunoassay. To validate the lateral flow immunoassay, the results were compared with HPLC-MS/MS (GC-MS/MS). The HPLC-MS/MS system was coupled to a TSQ Quantum Ultra EMR (Thermo Fisher Scientific, San Jose, CA, USA). The HPLC separation was conducted using a Thermo Scientific C_18_ column (Hypersil Gold, 100 mm × 2.1 mm, 3.0 μm) at 20 °C. The mass spectra were obtained by a triple quadrupole coupled with the electrospray interface (ESI). The MS/MS was operated using ESI sources in a positive or negative mode. The conditions were set as follows: spray voltage at 4.0 kV for ESI^+^ and −3.0 kV for ESI^−^, capillary temperature at 300 °C. The precursor ion and two product ions (quantification ion/qualification ion) for AFB_1_ (313.0 and 285.0), and ZEA (317.0 and 130.8) were recorded [38].

The GC-MS/MS system was coupled to a triple quadrupole (Shimadzu, Japan) in the selected multiple reactions monitoring (MRM) mode. The other parameters were column: DB-5ms capillary columns (30 m × 0.25 mm × 0.25 µm), carrier gas: high-purity He gas, column flow: 1 mL/min, PTV injector temperature: 65 °C, column procedure: 40 °C maintained for 4 min, 25 °C/min up to 125 °C, then 10 °C/min up to 300 °C maintained for 4 min. Ion source (EI) temperature: 250 °C; ionization voltage: 70 eV; the precursor ion and two product ions (quantification ion) for CTN (265.90 and 170.10) were also recorded.

### 4.8. Sample Preparation

Mycotoxin/pesticide-free and positive samples (maize, peanut) were prepared by spiking mixture standard solutions of AFB_1_/ZEA/CTN. The spiked samples were standing overnight at room temperature. A 10 g sample was homogenized for 2 min in 30 mL 70% methanol (*v*/*v*), followed by centrifugation at 8000× *g* for 2 min. After collecting the supernatants, 1 mL of supernatants was diluted with 3 mL 0.4% Tween-20 solution.

## Figures and Tables

**Figure 1 toxins-11-00056-f001:**
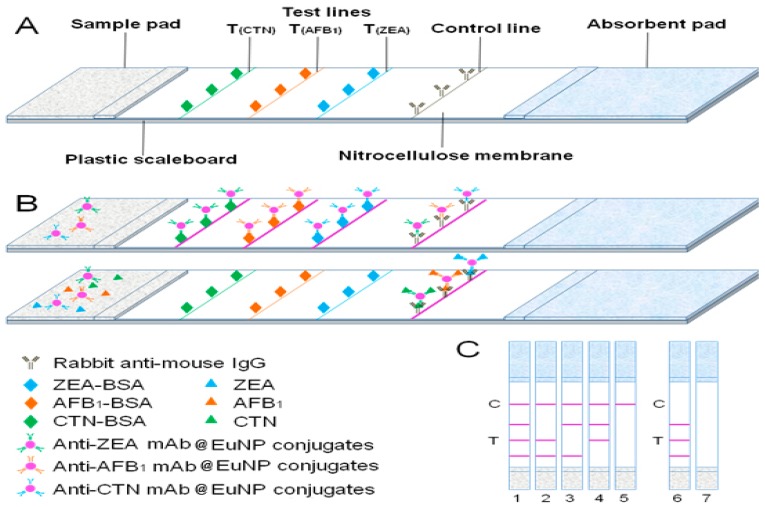
(**A**) Schematic of lateral flow immunoassay by time-resolved fluorescent immunochromatographic strip. (**B**) Immunoassay procedure for negative and positive samples, respectively. (**C**) Schematic illustration of the results 1) ZEA (−), AFB_1_ (−), CTN (−); 2) ZEA (+), AFB_1_ (−), CTN (−); 3) ZEA (−), AFB_1_ (+), CTN (−); 4) ZEA (−), AFB_1_ (−), CTN (+); 5) ZEA (+), AFB_1_ (+), CTN (+); 6) and 7) invalid. +, positive; −, negative.

**Figure 2 toxins-11-00056-f002:**
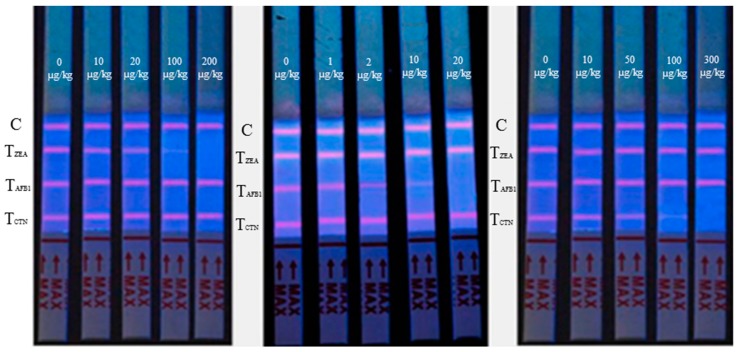
Specificity result spiked by series concentrations of AFB_1_, ZEA, and CTN, respectively. C: control line; T_ZEA_: test line (ZEA); T_AFB1_: test line (AFB_1_); T_CTN_: test line (CTN).

**Table 1 toxins-11-00056-t001:** Calibration curves for the analysis of ZEA, AFB_1_, and CTN.

Analyte	Matrix	Linear Equation	R^2^	LOD (µg/kg)	Linear Range (µg/kg)
AFB_1_	Maize	y = −765.2 log(x) + 1038.2	0.991	0.16	0.48–20.00
	Peanut	y = −743.9 log(x) + 1063.5	0.987	0.18	0.54–20.00
ZEA	Maize	y = −518.4 log(x) + 1354.1	0.990	0.52	1.56–200.00
	Peanut	y = −479.4 log(x) + 1325.5	0.984	0.57	1.71–200.00
CTN	Maize	y = −510.81 log(x) + 1532.4	0.991	1.21	3.63–300.00
	Peanut	y = −468.66 log(x) + 1479.2	0.983	1.47	4.41–300.00

**Table 2 toxins-11-00056-t002:** Recovery analysis for AFB_1_, ZEA, and CTN via spiked blank matrix.

Analyze	Sample	Spiked (µg/kg)	Inter-Day ^a^ (*n* = 5)		Intra-Day ^b^ (*n* = 5)	
			Found ^c^ (µg/kg)	Recovery (%)	Found (µg/kg)	Recovery (%)
AFB_1_	Maize	1.00	1.11 ± 0.11	110.80	1.08 ± 0.03	108.25
		5.00	4.64 ± 0.16	92.88	4.65 ± 0.03	92.96
		10.00	9.35 ± 0.20	93.48	9.42 ± 0.10	94.15
	Peanut	1.00	0.92 ± 0.08	91.60	0.96 ± 0.04	95.87
		5.00	4.61 ± 0.23	92.36	4.60 ± 0.04	91.91
		10.00	9.19 ± 0.25	91.90	9.18 ± 0.14	91.77
ZEA	Maize	5.00	4.43 ± 0.32	88.52	4.48 ± 0.05	89.60
		20.00	20.77 ± 0.96	103.83	20.67 ± 0.44	103.33
		60.00	55.28 ± 0.96	92.13	55.23 ± 0.45	92.05
	Peanut	5.00	4.33 ± 0.35	86.56	4.29 ± 0.04	85.72
		20.00	18.21 ± 0.95	91.07	18.18 ± 0.22	90.91
		60.00	53.63 ± 0.98	89.38	54.25 ± 0.58	90.41
CTN	Maize	10.00	8.63 ± 0.36	86.32	8.57 ± 0.28	85.66
		20.00	18.22 ± 0.76	91.11	18.07 ± 0.13	90.37
		50.00	44.14 ± 1.66	88.29	43.21 ± 0.82	86.43
	Peanut	10.00	8.41 ± 0.77	84.06	8.42 ± 0.16	84.16
		20.00	18.15 ± 0.85	90.74	17.92 ± 0.22	89.59
		50.00	41.62 ± 1.57	83.24	41.76 ± 0.40	83.52

^a^ Assays were performed among five replicates in the same day; ^b^ Assays were performed among five replicates on three different days; ^c^ Data were the SD from all repeated measured values.

**Table 3 toxins-11-00056-t003:** Comparison of the results for AFB_1_, ZEA, and CTN via lateral flow immunoassay and HPLC-MS/MS (or GC-MS/MS).

Sample	Number	HPLC		GC-MS/MS	Lateral Flow Immunoassay		
		AFB_1_ (µg/kg)	ZEA ^a^ (µg/kg)	CTN (µg/kg)	AFB_1_ (µg/kg)	ZEA (µg/kg)	CTN (µg/kg)
Maize	1	—	28.32 ± 1.16	27.60 ± 0.69	—	29.89 ± 2.33	24.38 ± 1.11
	2	31.53 ± 0.66	— ^b^	—	32.52 ± 2.29	—	—
	3	—	96.30 ± 1.47	—	—	87.98 ± 1.96	—
	4	—	67.93 ± 1.02	—	—	60.36 ± 2.09	—
	5	12.47 ± 0.46	—	31.51 ± 0.61	11.98 ± 0.93	—	29.54 ± 1.34
	6	—	142.29 ± 4.13	—	—	118.47 ± 3.11	—
	7	4.67 ± 0.38	37.35 ± 1.43	14.45 ± 0.44	5.20 ± 0.58	35.20 ± 2.06	17.87 ± 1.80
	8	45.79 ± 0.58	—	—	40.77 ± 1.51	—	—
Peanut	1	17.81 ± 0.51	—	47.85 ± 0.73	18.44 ± 0.81	—	36.69 ± 3.20
	2	29.82 ± 0.41	—	—	27.26 ± 1.02	—	—
	3	34.47 ± 0.44	—	22.41 ± 0.54	32.35 ± 0.64	—	20.42 ± 0.82
	4	6.71 ± 0.62	—	—	6.23 ± 0.57	—	—
	5	—	—	—	—	—	—
	6	54.62 ± 0.67	—	18.46 ± 0.60	48.41 ± 1.42	—	16.92 ± 0.78
	7	11.25 ± 0.40	—	—	10.83 ± 0.90	—	—
	8	—	—	—	—	—	—

^a^ Data are expressed as the mean ± SD of values from three repeated experiments; ^b^ “—” indicates not detected.

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
