# Peer review of "Simultaneous Lateral Flow Immunoassay for Multi-Class Chemical Contaminants in Maize and Peanut with One-Stop Sample Preparation"

_toxins, 2019, doi:10.3390/toxins11010056_

Round 1
Reviewer 1 Report
The article titled “Simultaneous point-of-care sensing method for multi-class chemical contaminants in maize and peanut with one-stop sample preparation” reports an interesting work for simultaneous determination of the three contaminants. The work is well constructed but the results are not properly presented. Before its possible publication in the Journal, some aspects have to be revised.
In Line 17, the acronym of gas chromatography-electron capture detection (GC-MS/MS) is erroneous.
In line 28, after description of fungi is written et al.
In line 64, et al. is written without sense again.
In line 85, the sentence “The pH value that affected the mAb coupling and stability of mAB-EuNp” is not sense.
Epigraphs 2.1 and 4.3 must be combined and clarified.
What the micropore is? Micropore is used to liophilizate the mAb-EuNP complex. I other sentence is written that “lyophilized mAb-EuNP conjugate was pippeted into micropores”. Extraction solution was added from the sample and after mixing, a strip was inserted… It would be nice to have a more clear explanation of this.
The epigraph 4.5 (Principle of POC sensing) is not clear at all. This must be combined with the first paragraph in epigraph 2.3.1 (calibration curve) and clarified.
In line 134, “The fluorescent intensities on the T and C lines were inversely proportional to the analysed concentration” it could be not true for the C line.
Figure 2 must be detailed.
TRFIA detection is not detailed at all. Furthermore, is difficult to consider this kind of detection as a POC system. It would be interesting for the authors to comment on this aspect.
Epigraphs 2.2 and 2.3 have the same description. In the optimization parameters, it would be interesting to show (perhaps as additional information) the data/graphs where these results are based on.
Epigraphs 2.4 and 4.8 must be combined. Authors claim that one-stop sample preparation for these three analytes is excepctional, but they are based on the same procedure.
In line 208, “When the CTN was present in the samples, neither AFB1 nor ZEA was detected” does not correlates with the table 3.
Author Response
The article titled “Simultaneous point-of-care sensing method for multi-class chemical contaminants in maize and peanut with one-stop sample preparation” reports an interesting work for simultaneous determination of the three contaminants. The work is well constructed but the results are not properly presented. Before its possible publication in the Journal, some aspects have to be revised.
l In Line 17,the acronym of gas chromatography-electron capture detection (GC-MS/MS) is erroneous.
Answer: Thanks for the comment. The sentence has been revised, please see in line 15-17,as the following:
An excellent correlation was observed between this approach and high-performance liquid chromatography-tandem mass spectrometry (HPLC-MS/MS) for mycotoxins and gas chromatography-tandem mass spectrometry (GC-MS/MS) for pesticide in maize and peanut.
l In line 28, after description of fungi is written et al.
Answer: Thanks for the comment. The sentence has been revised, please see in line 27-29,as the following:
Fungicides are commonly used to control adverse effect from fungal diseases, such as A. flavus, A. parasiticus, F. graminearum, F. culmorum and F. crookwellense in agricultural practices, affecting the nervous system, cancer, and causing acute death[5]
l In line 64,et al. is written without sense again.
Answer: Thanks for the comment. The sentence has been revised, please see in line 63-64,as the following:
The paper-based systems use a variety of labels, including gold, silver, fluorescent and electrochemical/magnetic particles.
l In line 85, the sentence “The pH value that affected the mAb coupling and stability of mAB-EuNp” is not sense.
Answer: Thanks for the comment. The sentence has been deleted, please see in line 85
l Epigraphs 2.1 and 4.3 must be combined and clarified.
What the micropore is? Micropore is used to liophilizate the mAb-EuNP complex. I other sentence is written that “lyophilized mAb-EuNP conjugate was pippeted into micropores”. Extraction solution was added from the sample and after mixing, a strip was inserted… It would be nice to have a more clear explanation of this.
Answer: Thanks for the comment. The sentence has been revised in the manuscript :
The micropore was changed to microwell.
l The epigraph 4.5 (Principle of POC sensing) is not clear at all. This must be combined with the first paragraph in epigraph 2.3.1 (calibration curve) and clarified.
Answer: Thanks for the comment. The epigraph 4.5 and epigraph 2.3.1 has been combined. Pleased see in epigraph 2.3.1:
2.3.1. Calibration curve
In order to establish a standard curve, the spiked sample and the standard solution were employed, respectively.For negative controls, the mAb@Eu-nanosphere particles present in the solution flowed laterally along the strip towards the T lines where the mAb@Eu-nanosphere particles complexes are captured by the immobilized antigens on the T lines [T(CTN), T(AFB1), T(ZEA)]. The excess mAb@Eu-nanosphere particles flowed towards the C line, and AFB1/ZEA/CTN-mAb@Eu-nanosphere particles conjugates were captured by the rabbit anti-mouse IgG on the C line. Then, the T and C lines were inspected by the multi-target reader. In the case of positive samples, AFB1/ZEA/CTN-conjugated mAb@Eu-nanosphere particles did not interact with the immobilized complex on the T lines, but reached the C line that was indicated by a faint red color (i.e., a positive sample with a low concentration of the analyte) or no color (i.e., a positive sample with a high concentration of the analyte) on the T lines (Figure 1B and 1C). These results showed that the fluorescent intensity was correlated with the concentration and the three T lines did not interfere with each other. The fluorescent intensities ratio on the T and C lines were inversely proportional to the analyzed concentration. Therefore, the levels of the analytes were determined in the samples
l In line 134, “The fluorescent intensities on the T and C lines were inversely proportional to the analysed concentration” it could be not true for the C line.
Answer: Thanks for the comment. The sentence has been revised, please see in line 132-133,as the following:
The ratio of fluorescent intensities on T and C lines were inversely proportional to the analyzed concentration.
l Figure 2 must be detailed.
Answer: Thanks for the comment. The Figure 2 has been detailed., please see in line 142-143,as the following:
The Series of concentrations for the analysis of AFB1 (0, 1, 2, 10, 20 µg/kg), ZEA (0, 10, 20, 100, 200 µg/kg), and CTN (0, 10, 50, 100, 300 µg/kg) were shown in Figure 2,
l TRFIA detection is not detailed at all. Furthermore, is difficult to consider this kind of detection as a POC system. It would be interesting for the authors to comment on this aspect.
Answer: Thanks for the comment. The POC system has been changed paper-based system. please see in the manuscript.
l Epigraphs 2.2 and 2.3 have the same description. In the optimization parameters, it would be interesting to show (perhaps as additional information) the data/graphs where these results are based on.
Answer: Thanks for the comment. The Epigraphs 2.2 and 2.3 has been combined and revised. please see in the manuscript.
l Epigraphs 2.4 and 4.8 must be combined. Authors claim that one-stop sample preparation for these three analytes is excepctional, but they are based on the same procedure.
Answer: Thanks for the comment. The sentence has been revised. please see in the Epigraphs 2.4 and 4.8. According to the principle of similar phase dissolution, the same method was used for pretreatment, and the recovery was evaluated, pleased see in the table 2.
l In line 208, “When the CTN was present in the samples, neither AFB1 nor ZEA was detected” does not correlates with the table 3.
Answer: Thanks for the comment. The sentence has been deleted, please see in line 202

Reviewer 2 Report
The experiments are decribed thoroughly and results are presented clearly. But there are some inconsistencies in the text which should be improved:
Line 12, 13: LOD and analytical range stated is for maize only (see table 1 on page 5). But it is not apparent from the context. The statement is made as it was valid for both maize and peanuts.
Line 27: Aspergillus flavus - in not disease but fungus. The same other for other fungi named (A.parasiticus, Fusarium….)
Line 33: aflatoxin B1 is not fungi but mycotoxin.
In the Introduction, some references should be replaced as they don´t support the given statement. E.g. References 9-11 are used to support the statement about toxic effects of mycotoxins. But papers 9-11 reffer to the development of different analytical methods. Similarly, the references 13 and 14.
Line 43-46: Number of European regulation limiting given mycotoxins and pesticide should be stated and overal, the paragraph devoted to limits should be improved. For example, the limit for zearalenone in unprocessed maize is 200μg/kg (in the text is 60μg/kg). Limits for aflatoxins in peanuts are missing but should be also involved, etc.
Abbreviations: Many abbreviations are explained in the part 4. Materials and methods but are currently used before, in the part 2. Results and discussion. There are many of them: EuNP, NC, …
Author Response
The experiments are decribed thoroughly and results are presented clearly. But there are some inconsistencies in the text which should be improved:
l Line 12, 13: LOD and analytical range stated is for maize only (see table 1 on page 5). But it is not apparent from the context. The statement is made as it was valid for both maize and peanuts.
Answer: Thanks for the comment. The sentence has been revised, please see in line 63-64,as the following:
At optimal conditions, the sensing method lowered the limits of detection (LOD) to 0.16, 0.52, and 1.21 µg/kg in maize and 0.18, 0.57, and 1.47 µg/kg in peanut with an analytical range of 0.48-20, 1.56-200, and 3.63-300 µg/kg for AFB1, ZEA and CTN, respectively.
l Line 27: Aspergillus flavus - in not disease but fungus. The same other for other fungi named (A.parasiticus, Fusarium….)
Answer: Thanks for the comment. The sentence has been revised, please see in line 11-14,as the following:
Fungicides are commonly used to control adverse effect from fungal diseases, such as A. flavus, A. parasiticus, F. graminearum, F. culmorum and F. crookwellense in agricultural practices, affecting the nervous system, cancer, and causing acute death.[
l Line 33: aflatoxin B1 is not fungi but mycotoxin.
Answer: Thanks for the comment. The sentence has been revised, please see in line 33-34,as the following:
Among the mycotoxin, aflatoxin B1 (AFB1) and zearalenone (ZEA) are frequently and widely found in agricultural products
l In the Introduction, some references should be replaced as they don´t support the given statement. E.g. References 9-11 are used to support the statement about toxic effects of mycotoxins. But papers 9-11 reffer to the development of different analytical methods. Similarly, the references 13 and 14.
Answer: Thanks for the comment. The sentence has been revised, please see in line 31-37,as the following:
On the other hand, mycotoxins are mainly produced by fungi, and are carcinogenic, teratogenic, nephrotoxic, and mutagenic. Among the mycotoxin, aflatoxin B1 (AFB1) and zearalenone (ZEA) are frequently and widely found in agricultural products.[10] AFB1, the Group 1 carcinogen classified by the International Agency for Research on Cancer, can lead to liver cirrhosis, carcinomas, and lethality in human and livestock [11].
l Line 43-46: Number of European regulation limiting given mycotoxins and pesticide should be stated and overal, the paragraph devoted to limits should be improved. For example, the limit for zearalenone in unprocessed maize is 200μg/kg (in the text is 60μg/kg). Limits for aflatoxins in peanuts are missing but should be also involved, etc.
Answer: Thanks for the comment. The sentence has been revised, please see in line 45-46,as the following:
The MRL for ZEA is 60 μg/kg in the unprocessed maize, AFB1 is 20 μg/kg, while the ones are 50 and 100 μg/kg for CTN in peanut and maize, respectively.
l Abbreviations: Many abbreviations are explained in the part 4. Materials and methods but are currently used before, in the part 2. Results and discussion. There are many of them: EuNP, NC,
Answer: Thanks for the comment. The abbreviations such as EuNP, NC has been revised, in the manuscript.
